# Does Treatment Readiness Shape Service-Design Preferences of Gay, Bisexual, and Other Men Who Have Sex with Men Who Use Crystal Methamphetamine? A Cross Sectional Study

**DOI:** 10.3390/ijerph19063458

**Published:** 2022-03-15

**Authors:** Kiffer G. Card, Madison McGuire, Graham W. Berlin, Gordon A. Wells, Karyn Fulcher, Tribesty Nguyen, Trevor A. Hart, Shayna Skakoon Sparling, Nathan J. Lachowsky

**Affiliations:** 1Faculty of Health Sciences, Simon Fraser University, Burnaby, BC V5A 1S6, Canada; 2School of Population and Global Health, McGill University, Montréal, QC H3A 0G4, Canada; madison.mcguire@mail.mcgill.ca; 3Department of Psychology, Ryerson University, Toronto, ON M5B 2K3, Canada; gberlin@ryerson.ca (G.W.B.); s.sparling@ryerson.ca (S.S.S.); 4School of Public Health and Social Policy, University of Victoria, Victoria, BC V8P 5C2, Canada; alexwells@uvic.ca (G.A.W.); kfulcher@uvic.ca (K.F.); nlachowsky@uvic.ca (N.J.L.); 5Faculty of Medicine, University of British Columbia, Vancouver, BC V6T 1Z4, Canada; tribesty.nguyen@alumni.ubc.ca

**Keywords:** methamphetamine, gay and bisexual men, patient-oriented care, intervention design, readiness to change

## Abstract

Crystal methamphetamine (CM) disproportionately impacts gay, bisexual, and other men who have sex with men (gbMSM). However, not all gbMSM are interested in changing their substance use. The present study aimed to examine whether participant-preferred service characteristics were associated with their readiness to change. We surveyed gbMSM who used CM in the past six months, aged 18 plus years, on dating platforms. Participants rated service-design characteristics from “very unimportant” to “very important”. Multivariable regression tested service preference ratings across levels of the Stages of Change Readiness and Treatment Eagerness Scale (SOCRATES-8D). Among 291 participants, 38.7% reported their CM use was not problematic, 19.5% were not ready to take any action to reduce or stop using CM, and 41.7% were ready to take action. On average, participants rated inclusive, culturally-appropriate, out-patient counselling-based interventions as most important. Participants with greater readiness-to-change scores rated characteristics higher than gbMSM with lesser readiness. Contingency management and non-abstinence programming were identified as characteristics that might engage those with lesser readiness. Services should account for differences in readiness-to-change. Programs that provide incentives and employ harm reduction principles are needed for individuals who may not be seeking to reduce or change their CM use.

## 1. Introduction

Globally, gay, bisexual, and other men who have sex with men (gbMSM) are considerably more likely to use crystal methamphetamine (CM) compared with straight-identified men [1]. Culturally-appropriate and low-barrier services are needed to help gbMSM manage their CM use and avoid harms that might arise from it, such as participation in behaviours that increase risk of HIV and other STI acquisition (e.g., unprotected sex and sharing needles when injecting drugs) [2]. In addition, CM has been found to be used by gbMSM as a way to cope with mental health issues and trauma in the absence of necessary mental health and social supports [3]. To meet this need, a variety of psychosocial, pharmacological, and harm reduction interventions have been designed and tested to assist gbMSM with CM cessation or reduction [4]. These interventions have been shown to reduce psychological distress and improve quality of life for participating gbMSM [5].

Unfortunately, many of these interventions require abstinence or have other stringent requirements for participation (e.g., frequent meetings or dogmatic foci) [4]. As a result, attrition rates for these programs can be as high as 70% or 80% in what are already typically small scale interventions serving only between 50 and 200 patients [4,6,7,8,9,10,11]. Program limitations pose a significant barrier to care among gbMSM who are not ready to abstain or who cannot adhere to rigorous programming [12].

Flexible program goals and requirements are important given that goals such as abstinence for gbMSM may not always be feasible for a variety of reasons, including the social and cultural context of CM use for many within this community [2,13]. Sex and sexuality are important components of social circles for many gbMSM (i.e., sex based sociality) and, given the positive effects of CM on sexual pleasure, endurance, and intimacy, it is often used in these settings. Since substance use patterns among gbMSM are often shaped by their social and sexual interactions, they may be less willing to completely abstain from using [14].

Patient-oriented approaches that meet participants ‘where they are’ may provide opportunities to improve health services for the diverse population of gbMSM who are at various stages of readiness to change their CM use [12,15]. A meta-analysis of seven studies with 723 participants demonstrated that culturally-appropriate treatments are associated with significantly larger reductions in post-treatment substance use levels compared with alternative conditions in which cultural considerations are not considered or emphasized [16]. Recognizing this, researchers have called for investment in implementation research that aims to understand what types of interventions and what program characteristics are most effective and best received by people who use CM [17]. Given the interrelated dynamics of sexual activity, identity, sociality, and CM use, such investment is especially important for gbMSM [18].

There is a clear justification for identifying the program characteristics preferred by gbMSM who use CM. A study of 20 people who used CM showed that internal motivations play a critical role in shaping engagement and success in CM interventions [19,20]. A systematic review of barriers to substance use treatment also showed that problem recognition, treatment readiness, and fear of stigma and discrimination were key factors shaping help-seeking behaviour among people with substance use disorders [21]. These studies support readiness for change as a key factor in understanding substance use care for people who use drugs.

The overall goal of the present study was to understand the role of readiness to change in shaping the program preferences of gbMSM who use CM. Understanding how preferences are shaped by a participant’s readiness to change their substance use can help interventionists develop supports and services that are acceptable to those who are ready to engage in treatment, broadly defined. Further, understanding what interventions are acceptable by people who are not yet ready to change can support the implementation with an audience less likely to engage themselves in current treatment options. With this in mind, the present study aimed to (1) identify preferred service design characteristics of gbMSM who used CM and (2) assess whether these preferences were associated with treatment readiness. We hypothesized that individuals with higher readiness for change (as measured using the Stages of Change Readiness and Treatment Eagerness Scale (SOCRATES) [22] would be more likely to take a positive view of potential program characteristics—rating them higher and being more willing to participate in more intensive interventions. This hypothesis is based on an underlying assumption that the program characteristics valued by participants are dependent on their readiness to change and willingness to engage in care. Thus, accounting for change readiness can help us better tailor interventions to individuals at different stages of readiness and at a population-level to better predict service needs by target audience segment/size—particularly with respect to the kinds of services individuals may choose according to their readiness to change. Such an approach is supported by the growing body of evidence examining the benefits of harm reduction interventions among people who use drugs [23,24].

## 2. Material and Methods

### 2.1. Protocol

We recruited participants primarily through advertisements on Squirt and Scruff (Figure 1), which are online dating/sex-seeking sites, and social media posts shared by the research team and community partners (e.g., Community-based Research Centre, Gay Men’s Sexual Health Alliance) on Facebook, Twitter, and Reddit. The first advertisements and promotions were piloted in British Columbia on 14 February 2020 using Squirt. In late March 2020, recruitment was expanded across Canada with a second Squirt e-blast on 27 March 2020. From 12 to 26 May 2020, a pop-up ad was promoted on the Scruff app. Potential participants who clicked on a study advertisement or social media post were directed to a web-based survey. Participants provided informed consent and were then screened for eligibility. Pre-determined eligibility criteria restricted participation to individuals 18 years of age or older who gender-identified as a man (inclusive of trans men) or as genderqueer/non-binary, and who had reported both sex with a man and CM use in the past six months. Eligible participants completed an English-language questionnaire that was developed based on qualitative interviews with gbMSM who used CM [25]. The questionnaire assessed a wide range of demographic, behavioural, and psychosocial variables and took between 30 and 45 min to complete. Upon completion of the survey questionnaire, participants had the option to provide their contact information to be sent an e-transfer or check for $10 CAD. Honoraria requests were monitored for duplicates. 

### 2.2. Variables

#### 2.2.1. Service Design Preferences

To assess preferred characteristics for substance-use services and supports participants were asked: “If you wanted to access professional help to control, cut down, or stop using crystal methamphetamine (crystal, meth, tina, etc.) how important would each of the following characteristics of the program be to you?” This question was followed by a list of 31 program characteristics that were developed based on previously conducted qualitative interviews and survey development consultations with stakeholder groups. Broadly, these characteristics related to staff (e.g., “The staff have experience using methamphetamine”), other program participants (e.g., “The other participants identify as LGBTQ2S”), host organization characteristics (e.g., “The program is run through an LGBTQ2S organization”), the intervention itself (e.g., “The program includes one-on-one counselling”), harm reduction (e.g., “The program does not require abstinence from methamphetamine to participate”), and added benefits of participation (e.g., “I am given money for participating in the program”). Participants rated each characteristic on a four-point Likert scale: (1) Very unimportant, (2) Somewhat unimportant, (3) Somewhat important, or (4) Very important. Participants also indicated preferences for the ideal number of weeks and sessions a program would take place over, the ideal number of hours of each session, and the frequency of meetings (Daily; Once every few days; Once a week; Once every two weeks; Once a month). Since treatment programs may initially start as research projects, we asked about willingness (Yes; No) to participate in a research study that would help them (a) Quit, (b) Control or cut down; or (c) Stop their crystal methamphetamine use. Participants were asked whether they would be willing to participate in a randomized placebo-controlled trial testing the efficacy of pharmacological interventions (Yes; No), how much they would need to be compensated in order to participate in such a trial, and what types of compensation would be acceptable (Cash; Gift card; Prize draw; Free safer sex supplies; Free drug use supplies; Free food). 

#### 2.2.2. Readiness to Change

To assess a participant’s readiness to change their substance use, participants completed the Stages of Change Readiness and Treatment Eagerness Scale (SOCRATES) [22], which has previously been shown to correlate with length of stay in treatment and successful completion of treatment [26,27,28,29]. The SOCRATES is a 19-item personal drug use questionnaire consisting of three subscales that measure contemplativeness (formerly “Ambivalence”; e.g., “Sometimes I wonder if I am addicted to drugs”), readiness (formerly “Recognition”; e.g., “I really want to make changes in my use of drugs”), and action-taking (formerly “Taking Steps”; e.g., “I am actively doing things now to cut down or stop my use of drugs”). Each item was rated using a five-point Likert Scale: (1) Strongly Disagree, (2) Disagree, (3) Unsure or undecided, (4) Agree, (5) Strongly Agree. Higher scores on the contemplativeness subscale (range: 4–20) represent an openness to reflection on the harms arising from substance use. Higher scores on the readiness subscale (range: 4–35) represent an acknowledgment of problems, a desire to change, and a recognition of harms that might arise if they do not change. Higher scores on the action-taking subscale (range: 8–40) represent the participant’s current level of effort to change their substance use. The SOCRATES subscales have previously been shown to demonstrate excellent internal consistency, test-retest reliability, and convergent validity with other psychiatric measures of readiness-to-change [30].

#### 2.2.3. Demographic and Behavioural Characteristics

Several socio-demographic and behavioural characteristics were included based on previous research showing that key demographic and behavioural subgroups of gbMSM respond more favourably to interventions and campaigns [31]. Variables controlled for in this analysis included age (in years), whether the participant identified as a person of colour (Yes; No), gender (cisgender; transgender/non-binary), whether the participant identified as gay (Yes; No), province of residence (Grouped due to some small cell counts as The Prairies (Alberta, Manitoba, and Saskatchewan); Eastern and Atlantic Canada (Ontario, Quebec, New Brunswick, Newfoundland and Labrador, Nova Scotia, Prince Edward Island); Western Canada (British Columbia and Yukon Territory)), and HIV-status (I am HIV-positive; I think I am HIV-negative/I have never been tested for HIV). To assess frequency of CM use, participants were asked: “In the past six months, have you used crystal methamphetamine (crystal, meth, tina, etc.)?” and were provided with five categorical responses: (0) Not in the past 6 months, (1) Once or Twice, (2) Monthly, (3) Weekly, (4) Daily or Almost Daily. Participants who indicated “Not in the Past Six Months” were not eligible for participation and exited the survey after completing the eligibility screener.

### 2.3. Data Analysis

All data analyses were completed in R Studio (R Studio Team, Version 1.3.1073). Given our use of multivariable regression modelling in this study (described below), a complete case analysis was performed [32]. Incomplete observations were omitted using the na.omit() function. Differences between included and excluded participants were identified using χ^2^ tests for categorical variables and Wilcoxon rank sum tests for non-normal numeric variables. These test were constructed using the chisq.test() and wilcox.test() functions, respectively. The summary() function was used to calculate the minimum, maximum, median, first quartile (Q_1_), and third quartile (Q_3_) values for numeric variables. This included the calculation of average ratings for program characteristics rated on a four-point Likert scale (i.e., “1—Very unimportant” to “4—Very important.”) The table() and prop.table() functions were used to calculate frequencies and proportions for categorical variables. The cronbach.alpha() function in the “Itm” package was used to calculate study α and bootstrapped 95% confidence intervals for the three SOCRATES subscales and for a full version of the scale. We opted to keep the three SOCRATES subscales separate despite the complexity this decision adds to the interpretation of study findings. This is consistent with the analytic approach of other published studies using the SOCRATES [33]. 

In addition to the descriptive statistics, a bar chart was constructed in Microsoft Excel showing participant ratings of the importance of 31 potential program characteristics. For the item assessing the importance of having participants of the same ethnicity, stratified descriptive statistics and a chi-square test were calculated to test whether preference for this characteristic differed between participants according to whether they were identified as a person of colour. 

To assess whether program preferences were dependent on treatment readiness and eagerness, we treated each service design characteristic as an outcome and each of the three SOCRATES subscales as a primary explanatory variable in separate regression models. Ordinal regression models were constructed using the polr() function in the “MASS” package for ordered categorical outcomes. Numeric outcomes were assessed for normality using the shapiro.test() function (i.e., Shapiro–Wilk test), and then analyzed using a generalized linear model (glm() function). Binary outcomes were analysed using a logistic regression model constructed using the glm() function. Odds ratios (OR) were calculated using exponentiated regression coefficients and 95% confidence intervals were constructed using the confint() function. All regression models controlled for demographic characteristics and frequency of CM use. 

### 2.4. Ethical Approval

Ethical review for this study was conducted through Research Ethics BC, which provided harmonized ethics review and approval from the University of British Columbia, Simon Fraser University, and the University of Victoria (protocol #BC17-485). 

## 3. Results

### 3.1. Sample Description

A total of 1154 surveys were initiated, 803 participants passed the eligibility screener and provided informed consent, and 291 participants provided answers on all variables included in this analysis. Significant participant attrition was observed throughout the demographic and substance use patterns section of the questionnaire, which appeared before the assessment of treatment preferences and SOCRATES scale. Excluded participants were not different from included participants based on ethnicity (*p* = 0.98), gender (*p* = 0.74), province (*p* = 0.99), HIV-status (*p* = 0.98), income-level (*p* = 0.30), patterns of CM frequency (*p* = 0.40), contemplativeness subscale scores (*p* = 0.22), or action-taking subscale scores (*p* = 0.09). Included participants, however, were more likely to identify as gay (79.4% vs. 69.9%, *p* = 0.01) and have higher readiness subscale scores (Mean = 20.35 vs. 19, *p* = 0.05) compared with excluded participants. The contemplativeness (study α = 0.87, 95% CI = 0.79–0.86), readiness (study α = 0.93, 95% CI = 0.92–0.94), and action-taking (study α = 0.91, 95% CI = 0.89–0.93) subscales of the SOCRATES all had good internal consistency and participants scored across the full possible range of each measure. Due to high internal consistency of the full scale (study α = 0.94, 95% CI = 0.93–0.95), we considered treating the SOCRATES score as a single scale. However, correlation coefficients calculated using the cor.test() function showed low correlation between the contemplativeness and action subscales (r = 0.58, *p* < 0.0001) and between the readiness and action subscales (r = 0.55, *p* < 0.0001). 

Table 1 provides an overview of sample demographics. The majority of the analysed sample was comprised of cisgender gay men. One-in-four participants identified as a person of colour. Most participants lived in British Columbia, Ontario, and Quebec. The median age of the sample was 41 years. One-third of participants were living with HIV and one-third used CM on a daily or almost daily basis. 

### 3.2. Preferred Program Characteristics and Association with SOCRATES Subscale Scores

Figure 2 shows participant’s ratings of potential program characteristics and Table 2 shows results from multivariable regression models examining the relationship between program preferences and each of the SOCRATES subscales. Among included participants, 38.7% reported their CM use was not problematic, 19.5% were not ready to take any action to reduce or stop using CM, and 41.7% were ready to take action. The median values for the contemplativeness, readiness, and action taking subscales were 12 (Q_1_–Q_3_: 9–15, Min = 7, Max = 35), 21 (Q_1_–Q_3_: 14–26, Min = 4, Max = 20), and 24 (Q_1_–Q_3_: 18–29, Min = 8, Max = 40). Table 2 also includes the average rating for each of the characteristics that were assessed in terms of importance. Overall, most program characteristics explored were rated as important or somewhat important by at least one-in-five participants. Furthermore, correlations between participant ratings and the SOCRATES subscales were generally small. The highest average effect sizes were seen on the contemplativeness subscale (average aOR = 1.07), followed by the readiness (average aOR = 1.05), and then action taking subscales (average aOR = 1.03). The paragraphs below summarize these results in greater detail, organized thematically. 

#### 3.2.1. Staff Characteristics

Several characteristics concerning program staff were among the highest rated program characteristics we assessed. In particular, participants reported that having staff who understood the role of drugs in their life (85%), mental health (82%), and identity (70%) were important, as was having staff who identify as LGBTQ2S+ (66%) or have lived experience using CM (74%). Having understanding program staff was positively associated with all three subscales of the SOCRATES, though having staff that identify as LGBTQ2S+ was only significantly associated with the contemplativeness subscale. Having staff with lived experience using CM was not correlated with SOCRATES subscale scores.

#### 3.2.2. Participant Characteristics

Regarding the characteristics of other program participants, having LGBTQ2S+ participants was rated as somewhat important or very important by a majority of the sample. Meanwhile, having program participants of the same ethnicity was among the lowest rated characteristics we examined; there was no statistical difference in the ratings between participants who did and did not identify as a person of colour (*p* = 0.1479): 17.3% of white participants and 28.9% of participants of colour rated this feature very or somewhat important. Having participants in a similar financial situation fell somewhere between these two other measures—with nearly 50% rating this as a somewhat (35%) or very (14%) important characteristic. None of these three measures were associated with SOCRATES subscale scores. For the most part, few participants felt it was somewhat (20%) or very (9%) important for the program to involve people close to the participant. However, 68% of participants said it was important for the program to give them opportunities to help other participants, 66% said it was important for the program to include social activities with other participants, and 62% said it was important to be able to make friends and build relationships as part of the program. Higher SOCRATES subscale scores were associated with higher ratings of the importance of these social characteristics.

#### 3.2.3. Host Organization Characteristics

Only two host organization characteristics were considered in the present study: 49% of participants felt that having the organization run through an LGBTQ2S+ organization was somewhat or very important. This characteristic was associated with higher contemplativeness subscale scores on the SOCRATES. Having a program close to home was rated as somewhat (47%) or very (26%) important by the majority of participants. This was independent of SOCRATES subscale scores.

#### 3.2.4. Privacy and Disclosure Characteristics

Most participants said it was very (43%) or somewhat (36%) important that they were able to express themselves. This characteristic was associated with higher SOCRATES subscale scores. Two-thirds (64%) of participants also thought their anonymity was important, though they did not think it was important that the entire program was conducted one-on-one or that they would not have to disclose their sexual orientation. Of these three characteristics, only the ability to not disclose one’s sexuality was associated with SOCRATES subscale scores (contemplativeness and action-taking subscales, but not the readiness subscale).

#### 3.2.5. Intervention Characteristics

A wide range of intervention characteristics were assessed in the present study. Briefly summarizing these, 82% thought it was somewhat or very important that the program include one-on-one counselling, 78% thought it was somewhat or very important to have counsellor-led group counselling, and 67% thought it was somewhat or very important to have peer-led group counselling. While having a residential program was only endorsed as very or somewhat important by 29% of participants, 41% thought it was important that the program offers a place to detox. Higher rated importance on each of these intervention characteristics was associated with higher SOCRATES subscale scores.

Participants broadly felt that elements of harm reduction should be included in the design of programs: 74% thought it was important that programs had long-term ongoing support with no set end date, 59% thought it was important that abstinence was not required, 57% thought it was important that they would be able to take other drugs and still participate in the program, and 43% thought it was important that harm reduction supplies be provided. The only harm reduction characteristic that was associated with SOCRATES subscale scores was the provision for long-term and ongoing support with no set end date.

#### 3.2.6. Ideal Session Time, Frequency, and Duration

The median preferred duration for intervention programs was 10 weeks (Q_1_–Q_3_ = 5–12) and 15 sessions (Q_1_–Q_3_ = 10–26). Higher SOCRATES subscale scores were associated with increased odds of endorsing longer term programs. Similarly, SOCRATES subscale scores were associated with a higher ideal session time commitment. Overall, 51.5% of respondents felt that sessions should last 1 h, 35.4% said sessions should last longer than 1 h, and 9.1% said sessions should last less than 1 h. Regarding frequency, 14.8% wanted daily sessions, 41.9% wanted semi-weekly sessions, 36.1% wanted sessions only once per week, and 9.1% wanted sessions held on a less than weekly basis. Increasing desired frequency was associated with higher SOCRATES subscale scores.

Pharmacological intervention components were also recognized as important, with 49% saying it is important to be prescribed another medication that can help with reducing CM use (e.g., withdrawal management), 33% of participants saying it was important they were prescribed antidepressants, 36% saying it was important they were prescribed anxiety medications. Higher ratings on the importance of these characteristics were associated with higher SOCRATES subscale scores.

#### 3.2.7. Benefits of Participation

The median desired honorarium for participating in an intervention was $50 CAD per visit (Q_1_–Q_3_ = 20–100). The preferred modes of compensation were cash (85.9%), gift card (38.1%) and food (15.8%). Only a small minority of participants wanted to be compensated by entering lotteries/draws for prizes (7.9%), or receiving harm reduction supplies for substance use (5.8%), or safer sex supplies (5.5%). Of note, all participants in this study were provided with an option to receive $10 cash honoraria. Increasing SOCRATES subscale scores were associated with a lower expected honorarium for participation. Expectation for receiving money as part of the program (i.e., contingency management) was not associated with any of the SOCRATES subscale scores.

#### 3.2.8. Willingness to Participate

Finally, most participants expressed willingness to participate in research-based interventions: 62.2% said they would participate in a study that aimed to help them quit CM, 72.5% said they would participate in a study that aimed to help them stop (temporarily) their use, and 77.0% said they would participate in a study that aimed to help them control their use of CM. A total of 75.6% of respondents also said they would be willing to participate in a placebo-controlled trial to test the efficacy of pharmacological interventions that can help people with their dependence on CM. Willingness to participate in each type of program was associated with higher SOCRATES subscale scores.

## 4. Discussion

### 4.1. Key Findings

In the present study we identified program design preferences of 291 gbMSM who used CM in the past six months—which is important to understand how gbMSM might utilize services that are designed to support their CM use. We also tested whether service-design preferences were associated with treatment readiness and eagerness, as measured using three SOCRATES subscales—which provides insight into how individuals with different experiences, cultures, and attitudes about CM use might utilize services or be incentivized to do so. To our knowledge, this is the first such paper examining the relationship between readiness to change and the treatment preferences of participants, though other studies have examined the impact of readiness on treatment engagement and success [26].

In conducting these analyses, we hypothesized that most treatment preferences would be rated highly and that participants with higher SOCRATES subscale scores would rate hypothetical program characteristics higher and be more willing to participate in longer, more frequent, and longer-term interventions. Based on our results we found that our hypotheses were generally supported. Of the 31 program characteristics we asked participants to rate, 19 were rated as either somewhat or very important by more than 50% of the sample. Six additional characteristics were rated as somewhat or very important by at least one-third of participants. Similarly, 20 of the 31 characteristics that were rated based on their importance were positively correlated with at least one of the SOCRATES subscales—and most were correlated with all three subscales. Furthermore, greater readiness was associated with greater odds of being willing to participate in interventions, regardless of whether they were designed to help individuals quit, control, or temporarily stop using CM. Participants with higher readiness scores reported that they preferred programs with longer session times, greater session frequency, and greater program duration. Those with higher SOCRATES subscale scores also had lower expectations for the honorarium amount they expected to receive for participating in a research-based intervention trial.

These results seem to suggest that willingness to invest more time and effort in treatment programs is associated with being more ready to change—an important finding in the context of existing systems that frequently result in coerced treatment of people who use CM [34]. However, these individuals also rated the importance of program characteristics more highly—perhaps suggesting they are more aware of the potential barriers and hurdles associated with treatment. Thus, they rated characteristics as more important based on the belief that these characteristics will help them achieve their goals.

### 4.2. Implications

Our results speak to practical program design characteristics. Generally, there is a greater preference for out-patient compared with in-patient residential treatment (which was only endorsed by 8% of participants as being very important). This finding agrees with the broader literature on preferences regarding inpatient and out-patient care. For example, a study of 137 people who use stimulant drugs showed that intensive out-patient treatments and psychotherapy are highly preferred to residential programs [35]. Regarding out-patient programs, participants in the current study supported, on average, programs that ran approximately 10 weeks and consisted of 15 one hour-long daily or semi-weekly sessions. However, it was also important to note that a plurality of participants endorsed the importance of having ongoing supports with no set end-date. This may suggest that treatment programs should consist of multiple stages—perhaps an intensive daily or semi-weekly program with ongoing social and peer led supports thereafter. This is supported by participants’ self-rated importance of characteristics, which show that interventions should include multiple forms of counselling (e.g., one-on-one, peer-led group, and counsellor-led group) that allow them the opportunity to express themselves, support others, and build relationships with other participants.

In addition to the design of interventions, our results also speak to the importance of the interpersonal dynamics of these interventions. Indeed, while participants demonstrated preferences for staff with lived experience or a high degree of understanding of the role that CM use played in their life, identity, and mental health, it was relatively less important that participants and staff shared identity markers such as socioeconomic status, ethnicity, or sexual orientation (though these were still important to a significant proportion of participants). Knowing that LGBTQ2S+ people face many barriers to care beyond those experienced by the general population, it is not surprising to see such a large proportion of participants say that it is important to have staff with a high degree of understanding and lived experience [36]. This is also important to consider given the social and cultural aspects of CM use within the gbMSM community [37,38]. Thus, when working with gbMSM, lived experience and understanding must encompass CM and the cultural and social context in which CM is commonly used [39]. With this in mind, it is also important to recognize that participants did not think it was important that programs allowed them to keep their sexuality anonymous, nor did they generally think it was important that the programs were entirely anonymous. Although our survey did not ask participants to express which traits were undesirable, the low ratings of importance on these characteristics (e.g., non-disclosure, anonymity) may suggest that participants want programs where they can be open about their sexuality. Given this reality, if intervention programs for gbMSM are combined with those working with other populations these need to be fully LGBTQ2S-affirming [4,17], though it should be noted that peer counselling for gbMSM may require a different lens of lived experience. Creating culturally-appropriate environments therefore appears to be a fundamental requirement for creating spaces where gbMSM can talk comfortably and openly about sexualized methamphetamine use [40,41].

Finally, our analyses highlight several potential barriers in engaging participants with lesser readiness to change. While a number of highly rated characteristics were identified as important for individuals regardless of readiness scores, our results showed that participants who were less ready to change reported that a wide array of program characteristics were unimportant to them. Considering small effect size estimates, this does not necessarily mean they are unwilling to participate in interventions, rather it could suggest that they have simply not considered the various barriers that might make participating in an intervention difficult, or what they might prefer in an intervention. More research is needed to understand what program characteristics are potentially of importance to people who are less ready to change—especially with attention to harm reduction that may support safer use of substances and limit harmful effects, such as HIV transmission.

In the present study, the inclusion of contingency management (i.e., providing participants with motivational incentives and tangible rewards if they achieve goals, such as abstinence) may be useful for reaching gbMSM less ready to change. Contingency management has been widely found to be effective at improving treatment outcomes [42,43,44], and, with a notable exception [45], studies have found that contingency management is an efficacious treatment for reducing CM use among gbMSM [46,47,48]. Additionally, the expected per session honoraria amount was the only characteristic negatively associated with SOCRATES subscale scores—suggesting that people with lower readiness wanted to get paid more for participating in an intervention or treatment program. Additionally, many participants, regardless of readiness scores, thought that harm reduction design features were important characteristics of interventions and most wanted programs that did not require abstinence. The abstinence requirement of programs is regularly identified as a key barrier to harm reduction for gbMSM who use CM [4,17,20].

### 4.3. Limitations

As with many substance use research studies, this project is vulnerable to several sources of bias and error [49]. Our sampling strategy—which targeted geosocial networking applications popular for sexualized substance use—is subject to bias from non-coverage (i.e., not all gbMSM use these apps), and non-response bias. Indeed, a recent systematic review of non-probability samples of this population indicated greater prevalence of substance use and higher socioeconomic status [50]. Future research studies may be able to validate our findings in a sample recruited using specialized recruitment strategies for less-engaged populations, such as Respondent-Driven Sampling. However, we note that in addition to the difficulty of recruiting a representative population of gbMSM who use CM, we find that there are high levels of non-response across survey items—particularly as the number of questions increase. This is indicative of the challenge and importance of survey design that might be easier to complete for target populations. For example, shorter questionnaires, less cognitively demanding questions, and higher honoraria might improve response rates allowing for more participants to be included at each analytic stage. In addition to challenges with data collection among this population, our measures also pose some difficulties. For example, in asking participants to rate the importance of specific program characteristics, we did not ask them to indicate when a program characteristic is not only unimportant, but not preferred. Innovation in the ways program preferences are assessed can improve measurement and provide insights not fully addressed here. We also note that with tests across three scales, there may be an elevated risk for errant findings. Caution should be taken and confidence intervals should be carefully reviewed. Replication in future studies is also warranted.

## 5. Conclusions

The present study highlighted a variety of design considerations that could benefit programs tailored to gbMSM who use CM. In particular, staff empathy, peer-engagement, and out-patient counselling interventions are key characteristics endorsed as important by most gbMSM included in this study. While further research is needed to understand better the treatment preferences of gbMSM, intervention design should account for differences in readiness to change—potentially by providing participation cash incentives and services that include harm reduction. Such full-spectrum services could improve gbMSM’s uptake and adherence to interventions for CM.

## Figures and Tables

**Figure 1 ijerph-19-03458-f001:**
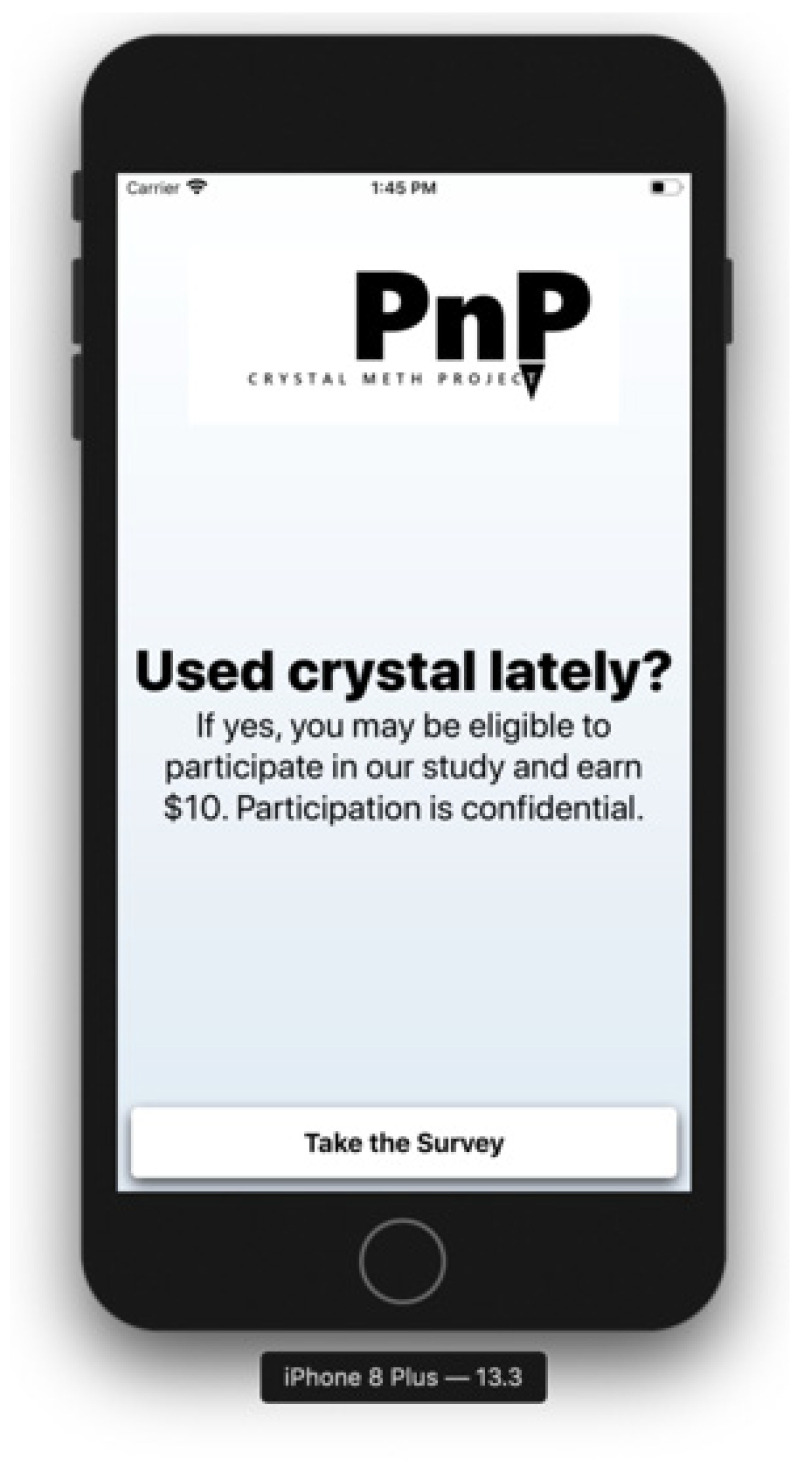
Example Ad used in recruitment.

**Figure 2 ijerph-19-03458-f002:**
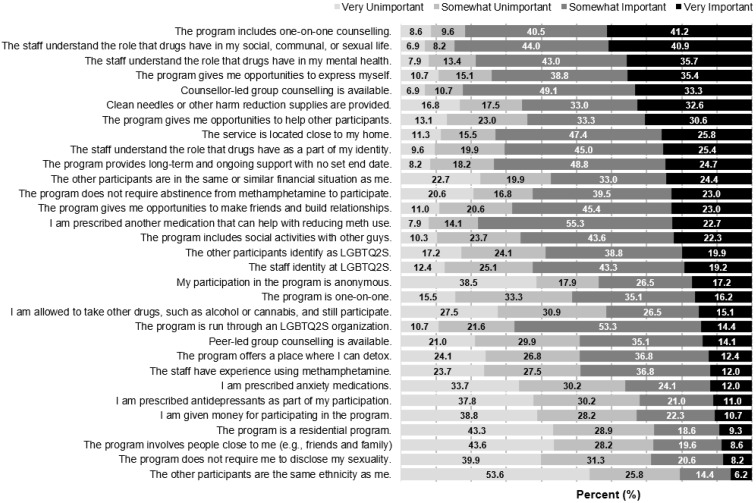
Participant’s Ratings of Potential Program Characteristics, ordered by Average Participant Rated Importance.

**Table 1 ijerph-19-03458-t001:** Sample Demographic Characteristics.

Variable	n (%)/*Median* (*Q*_1_–*Q*_3_)
**Age**	*41 (34–51)*
**Person of Colour**	
Yes	83 (28.5)
No	208 (71.5)
**Gender**	
Cisgender	269 (92.4)
Trans/Non-binary	22 (7.6)
**Sexual Identity**	
Gay	231 (79.4)
Bisexual	34 (11.7)
Other	26 (8.9)
**Province**	
Atlantic and Eastern Canada	155 (53.3)
The Prairies	32 (11.0)
Western Canada	104 (35.7)
**Person Living with HIV**	
Yes	107 (36.8)
No	184 (63.2)
**Annual Income (CAD)**	
Less than $29,999	100 (34.4)
$30,000–$59,000	99 (34.0)
$60,000–$89,999	52 (17.9)
$90,000 or more	40 (13.7)
**Frequency of CM Use in P6M**	
Daily or Almost Daily	102 (35.1)
Weekly	48 (16.5)
Monthly	49 (16.8)
Once or Twice	92 (31.6)
**Self-reported Readiness to take action on CM Use**	
No perceived problem with CM Use	113 (38.7)
Not ready to take action on CM Use	57 (19.5)
Ready to take action on CM Use	291 (41.7)
**SOCRATES Subscales**	
Contemplativeness (Range 4–20)	*12 (9–15)*
Readiness (Range 4–35)	*21 (14–26)*
Action-taking (Range 8–40)	*24 (18–29)*

Note: CM Use = Crystal Methamphetamine Use; P6M = Past Six Months; SOCRATES = Stages of Change Readiness and Treatment Eagerness Scale.

**Table 2 ijerph-19-03458-t002:** Multivariable Models Examining Associations between Program Design Preferences and SOCRATES Subscale Scores.

Outcome Variable	Average Rating (1–4)	Primary Explanatory Variable		
		Contemplativeness	Readiness	Action-Taking
		aOR (95%CI)	aOR (95%CI)	aOR (95%CI)
**Staff Characteristics**				
The staff identity as LGBTQ2S. ^A^	2.62	1.07 (1.01, 1.13)	1.02 (0.99, 1.06)	1.02 (0.99, 1.05)
The staff have experience using methamphetamine. ^A^	2.22	1.03 (0.97, 1.09)	1.02 (0.99, 1.06)	1.02 (0.99, 1.05)
The staff understand the role that drugs have in my social, communal, or sexual life. ^A^	3.14	1.09 (1.03, 1.16)	1.06 (1.02, 1.10)	1.06 (1.02, 1.09)
The staff understand the role that drugs have as a part of my identity. ^A^	2.86	1.11 (1.05, 1.18)	1.07 (1.03, 1.10)	1.05 (1.02, 1.09)
The staff understand the role that drugs have in my mental health. ^A^	3.09	1.10 (1.03, 1.16)	1.09 (1.05, 1.12)	1.04 (1.01, 1.07)
**Program Participant Characteristics**				
The other participants are in the same or similar financial situation as me. ^A^	2.81	1.04 (0.98, 1.10)	1.01 (0.98, 1.04)	0.98 (0.95, 1.01)
The other participants are the same ethnicity as me. ^A^	1.73	0.99 (0.94, 1.05)	0.99 (0.96, 1.03)	1.00 (0.97, 1.03)
The other participants identify as LGBTQ2S. ^A^	2.65	1.04 (0.98, 1.10)	1.01 (0.98, 1.05)	1.01 (0.98, 1.05)
The program involves people close to me. ^A^	1.94	1.06 (1.00, 1.12)	1.03 (1.00, 1.07)	1.02 (1.00, 1.05)
The program includes social activities with other guys. ^A^	2.69	1.08 (1.02, 1.15)	1.04 (1.01, 1.07)	1.04 (1.01, 1.07)
The program gives me opportunities to help other participants. ^A^	2.90	1.07 (1.01, 1.13)	1.06 (1.03, 1.10)	1.04 (1.01, 1.07)
The program gives me opportunities to make friends and build relationships. ^A^	2.78	1.14 (1.08, 1.21)	1.08 (1.05, 1.12)	1.07 (1.03, 1.10)
**Host Organization Characteristics**				
The program is run through an LGBTQ2S organization. ^A^	2.37	1.07 (1.01, 1.13)	1.02 (0.98, 1.05)	1.00 (0.97, 1.03)
The service is located close to my home. ^A^	2.88	1.05 (0.99, 1.11)	1.02 (0.98, 1.05)	1.02 (0.99, 1.05)
**Privacy and Disclosure Characteristics**				
My participation in the program is anonymous. ^A^	2.59	1.04 (0.99, 1.10)	1.03 (0.99, 1.06)	1.01 (0.99, 1.04)
The program does not require me to disclose my sexuality. ^A^	1.93	1.06 (1.00, 1.12)	1.00 (0.97, 1.03)	1.03 (1.00, 1.06)
The program is one-on-one. ^A^	2.52	1.05 (0.99, 1.11)	1.02 (0.99, 1.05)	1.01 (0.98, 1.04)
**Intervention Characteristics**				
The program gives me opportunities to express myself. ^A^	3.07	1.07 (1.01, 1.13)	1.05 (1.02, 1.09)	1.03 (1.00, 1.06)
The program is a residential program. ^A^	1.97	1.12 (1.06, 1.19)	1.08 (1.05, 1.12)	1.03 (1.00, 1.07)
The program offers a place where I can detox. ^A^	2.29	1.09 (1.03, 1.15)	1.07 (1.03, 1.10)	1.02 (1.00, 1.05)
The program includes one-on-one counselling. ^A^	3.19	1.13 (1.07, 1.20)	1.08 (1.05, 1.12)	1.04 (1.01, 1.08)
Counsellor-led group counselling is available. ^A^	2.99	1.10 (1.04, 1.16)	1.07 (1.04, 1.11)	1.04 (1.01, 1.07)
Peer-led group counselling is available. ^A^	2.37	1.10 (1.04, 1.17)	1.07 (1.03, 1.11)	1.06 (1.03, 1.10)
**Harm Reduction Characteristics**				
The program provides long-term and ongoing support with no set end date. ^A^	2.81	1.12 (1.05, 1.18)	1.07 (1.04, 1.11)	1.04 (1.01, 1.07)
The program does not require abstinence from methamphetamine to participate. ^A^	2.80	1.00 (0.95, 1.06)	0.99 (0.96, 1.03)	1.00 (0.97, 1.03)
I am allowed to take other drugs, such as alcohol or cannabis, and still participate. ^A^	2.42	0.98 (0.93, 1.04)	0.98 (0.95, 1.02)	0.99 (0.96, 1.02)
Clean needles or other harm reduction supplies are provided. ^A^	2.93	1.04 (0.99, 1.10)	1.01 (0.98, 1.04)	1.01 (0.98, 1.04)
**Ideal Session Time, Frequency, and Duration**				
Program Duration (In Number of Sessions) ^B^	-	1.01 (1.00, 1.02)	1.02 (1.02, 1.03)	1.01 (1.01, 1.02)
Program Duration (In Weeks) ^B^	-	1.01 (1.00, 1.02)	1.01 (1.01, 1.02)	1.02 (1.01, 1.02)
Session Time Commitment (In Hours) ^A^	-	1.11 (1.05, 1.17)	1.06 (1.02, 1.09)	1.06 (1.02, 1.09)
Session Frequency ^A^	-	1.05 (1.00, 1.12)	1.08 (1.04, 1.11)	1.03 (1.00, 1.06)
**Ideal Benefits of Participation**				
Session Honorarium Amount (CAD/visit) ^B^	-	0.95 (0.95, 0.96)	0.99 (0.99, 0.99)	0.97 (0.97, 0.97)
I am given money for participating in the program. ^A^	2.05	0.99 (0.93, 1.04)	1.00 (0.96, 1.03)	1.00 (0.97, 1.03)
I am prescribed antidepressants as part of my participation. ^A^	2.05	1.11 (1.05, 1.18)	1.06 (1.03, 1.10)	1.04 (1.01, 1.07)
I am prescribed anxiety medications as part of my participation. ^A^	2.14	1.08 (1.02, 1.14)	1.07 (1.04, 1.11)	1.03 (1.00, 1.06)
I am prescribed another medication that can help with reducing MA use. ^A^	2.71	1.10 (1.04, 1.17)	1.07 (1.03, 1.10)	1.04 (1.01, 1.07)
**Willingness to Participate in a…**				
Program Designed to Help Participants Quit using MA ^C^	-	1.20 (1.12, 1.28)	1.13 (1.09, 1.19)	1.10 (1.06, 1.14)
Program Designed to Help Participants Control MA use ^C^	-	1.17 (1.09, 1.27)	1.12 (1.06, 1.18)	1.08 (1.04, 1.12)
Program Designed to Help Participants Stop using MA Temporarily ^C^	-	1.18 (1.10, 1.27)	1.14 (1.09, 1.20)	1.10 (1.06, 1.14)
In a Placebo-Controlled RCT testing a Pharmacological Intervention ^C^	-	1.16 (1.08, 1.25)	1.09 (1.05, 1.15)	1.06 (1.02, 1.10)

Note: aOR = Adjusted Odds Ratios—All models control for age, ethnicity, gender, orientation, province, HIV status, income, and frequency of crystal methamphetamine use; 95%CI = 95% Confidence Interval; ^A^ = Ordinal Regression; ^B^ = Poisson Regression; ^C^ = Binary Logistic Regression.

## Data Availability

Data is available by request to the Authors.

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
