# Peer review of "Does Treatment Readiness Shape Service-Design Preferences of Gay, Bisexual, and Other Men Who Have Sex with Men Who Use Crystal Methamphetamine? A Cross Sectional Study"

_ijerph, 2022, doi:10.3390/ijerph19063458_

Round 1

Reviewer 1 Report

Excellent well-constructed and well-argued paper, interesting and very useful for those involved in prevention and treatment.

My only observation, minor, on the methodological level relates to the high proportion of missing data. This led to the size of the useful sample being reduced to approximately one third of the initial sample.

First of all, this point deserves to be mentioned in the "limitations" section, and be specified in particular the variables mainly concerned by non-responses in the “methods” section.

Then, to minimize this impact, the analysis could have been done in two stages:

1) analysis of participant's ratings of potential program characteristics on the entire sample integrating no-answer item) ;

2) Multifactorial regressions limiting to program characteristics that were rated as somewhat or very important by at least one-third of participants.

But this last remark is only a suggestion.

Author Response

REVIEWER 1

  • My only observation, minor, on the methodological level relates to the high proportion of missing data. This led to the size of the useful sample being reduced to approximately one third of the initial sample. First of all, this point deserves to be mentioned in the "limitations" section, and be specified in particular the variables mainly concerned by non-responses in the “methods” section. Then, to minimize this impact, the analysis could have been done in two stages: 1) analysis of participant's ratings of potential program characteristics on the entire sample integrating no-answer item); 2) Multifactorial regressions limiting to program characteristics that were rated as somewhat or very important by at least one-third of participants. But this last remark is only a suggestion.

Response: Thank you for this note and your suggestion. In exploring whether your suggested analysis plan might be helpful, we found that the missing data were not primarily on the program characteristics as these were all grouped closely together in the survey their patterns of non-response were similar. As such, we have noted this limitation and provided a bit of discussion to address this comment and R3.1, as follows:

As with many substance use research studies, this project is vulnerable to several sources of bias and error (Johnson, 2014). Our sampling strategy – which targeted geosocial networking applications popular for sexualized substance use – is subject to bias from non-coverage (i.e., not all gbMSM use these apps) and non-response bias. Indeed, a recent systematic review of non-probability samples of this population indicated greater prevalence of substance use and higher socioeconomic status (Salway et al., 2019). Future research studies may be able to validate our findings in a sample recruited using specialized recruitment strategies for less-engaged populations, such as Respondent-Driven Sampling. However, we note that in addition to the difficulty of recruiting a representative population of gbMSM who use CM, we find that there are high levels of non-response across survey items – particularly as the number of questions increase. This is indicative of the challenge and importance of survey design that might be easier to complete for target populations. For example, shorter questionnaires, less cognitively demanding questions, and higher honoraria might improve response rates allowing for more participants to be included at each analytic stage.

Reviewer 2 Report

Thank you for the opportunity of review this study entitled “Does Treatment Readiness Shape Service-Design Preferences of Gay, Bisexual, and Other Men Who Have Sex with Men Who Use Crystal Methamphetamine?: A Cross Sectional Study” (ijerph-1591813).

The study proposed an investigation about the role of readiness to change in shaping the program preferences of gbMSM who use CM. The research involved a sample of 291 gbMSM who used CM in the past six months.

In my opinion, the research topic is relevant, and the study is interesting. The paper is well written, logical, fluent, pleasant to read. Parallelly, there are some minor issues that need to be addressed before the paper will be suitable for publication.

  1. In the title, the double punctuation "?:" does not seem correct to me. I would eliminate the colon.
  2. Please remove the headings from the abstract, in line with the journal's guidelines.
  3. Please state the objectives of this study more clearly in the abstract, to offer a clear picture of what will be covered in the paper.
  4. Please reformat the references in line with the guidelines of the journal (Vancouver style).
  5. Please move the indications of internal consistency concerning the sample of the present study for the instruments that have been used from the data analysis section to the ones concerning the scales (In the “2.2 Variables” paragraph).
  6. The concluding sections are complete and well written. I really enjoyed this paper.

Author Response

REVIEWER 2

  • In the title, the double punctuation "?:" does not seem correct to me. I would eliminate the colon.

Response: As suggested, we removed the colon.

  • Please remove the headings from the abstract, in line with the journal's guidelines.

Response: As suggested, we have removed the section headings from the abstract.

  • Please state the objectives of this study more clearly in the abstract, to offer a clear picture of what will be covered in the paper.

Response: Within the limited word count we have tried to address this with an explicit statement of our aim:

Crystal methamphetamine (CM) disproportionately impacts gay, bisexual, and other men who have sex with men (gbMSM). However, not all gbMSM are interested in changing their substance use. The present study aimed to examine whether participant preferred service characteristics were associated with their readiness to change. We surveyed gbMSM who used CM in the past six months, aged 18+, on dating platforms. Participants rated service-design characteristics from “Very unimportant” to “Very important”.  Multivariable regression tested service preference ratings across levels of the Stages of Change Readiness and Treatment Eagerness Scale (SOCRATES-8D). Among 291 participants, 38.7% reported their CM use was not problematic, 19.5% were not ready to take any action to reduce or stop using CM, and 41.7% were ready to take action. On average, participants rated inclusive, culturally-appropriate out-patient counselling-based interventions as most important. Participants with greater readiness to change scores rated characteristics higher than gbMSM with lesser readiness. Contingency management and non-abstinence programming were identified as characteristics that might engage those with lesser readiness. Services should account for differences in readiness to change. Programs that provide incentives and employ harm reduction principles are needed for individuals who may not be seeking to reduce or change their CM use.

  • Please reformat the references in line with the guidelines of the journal (Vancouver style).

Response: We have changed the manuscript reverence to Vancouver Style.

  • Please move the indications of internal consistency concerning the sample of the present study for the instruments that have been used from the data analysis section to the ones concerning the scales (In the “2.2 Variables” paragraph).

Response: We have moved these sample descriptives to the results section.

  • The concluding sections are complete and well written. I really enjoyed this paper.

Response: Thank you for this compliment.  We are glad that you enjoyed reading the paper and found it to be well written.

Reviewer 3 Report

The authors report data from a study of readiness to change and service design preferences amongst MSM who have used crystal methamphetamine in the last six months.

Collecting valid data from this group can be especially challenging.  One indicator of this challenge is evidenced by the high attrition amongst those who actually did engage with the survey.  So not only is it difficult to recruit into the study but then to get complete data is another important challenge.

I did wonder whether the authors would like to comment or provide advice about future surveys of this type.  I'm interested in their thoughts about whether they tried to collect too much data and hence had survey that was too long?  What I mean is that an internet survey that runs for 30-45 mins for a $10CAD reimbursement is quite an investment of time by the participant.  I myself have been known to drop out of internet surveys where they take too long.  I'm curious about whether future surveys with a more targeted number of questions may result in larger sample size and less attrition.  Surveys such as these can suffer from being to small with potential for response bias.  I do acknowledge the work they did comparing the responders vs non-responders which was an important inclusion.

Overall, I was happy with the presentation of the data tables and figures and the accompanying interpretation.

Minor specific comments

Table 1.  Including range for the SOCRATES Subscales in the variable column of Table 1 may assist readers who are unfamiliar with SOCRATES to interpret these median scores.

Q1-Q3 incorrect for readiness - see below highlighted

"The median values for the contemplativeness, readiness, and action taking subscales were 12 (Q1-Q3: 9-15, Min = 7, Max = 35), 21 (Q1-Q3: 914-26, Min = 4, Max = 20), and 24 (Q1-Q3: 18-29, Min = 8, Max = 40)"

Table 2

The staff identity at LGBTQ2S. Should be "as"

Author Response

REVIEWER 3

  • Collecting valid data from this group can be especially challenging.  One indicator of this challenge is evidenced by the high attrition amongst those who actually did engage with the survey.  So not only is it difficult to recruit into the study but then to get complete data is another important challenge. I did wonder whether the authors would like to comment or provide advice about future surveys of this type.  I'm interested in their thoughts about whether they tried to collect too much data and hence had survey that was too long?  What I mean is that an internet survey that runs for 30-45 mins for a $10CAD reimbursement is quite an investment of time by the participant.  I myself have been known to drop out of internet surveys where they take too long.  I'm curious about whether future surveys with a more targeted number of questions may result in larger sample size and less attrition.  Surveys such as these can suffer from being to small with potential for response bias.  I do acknowledge the work they did comparing the responders vs non-responders which was an important inclusion.

Response: Thank you for this comment. We are glad that you found our comparison of responders and non-responders helpful. We also appreciate your comments regarding the difficulty of research with this population. We have integrated it into our response to R1.1, as shown below:

As with many substance use research studies, this project is vulnerable to several sources of bias and error (Johnson, 2014). Our sampling strategy – which targeted geosocial networking applications popular for sexualized substance use – is subject to bias from non-coverage (i.e., not all gbMSM use these apps) and non-response bias. Indeed, a recent systematic review of non-probability samples of this population indicated greater prevalence of substance use and higher socioeconomic status (Salway et al., 2019). Future research studies may be able to validate our findings in a sample recruited using specialized recruitment strategies for less-engaged populations, such as Respondent-Driven Sampling. However, we note that in addition to the difficulty of recruiting a representative population of gbMSM who use CM, we find that there are high levels of non-response across survey items – particularly as the number of questions increase. This is indicative of the challenge and importance of survey design that might be easier to complete for target populations. For example, shorter questionnaires, less cognitively demanding questions, and higher honoraria might improve response rates allowing for more participants to be included at each analytic stage

  • Table 1.  Including range for the SOCRATES Subscales in the variable column of Table 1 may assist readers who are unfamiliar with SOCRATES to interpret these median scores.

Response: Thank you for this suggestion. We have added the ranges to Table 1 in the first column, as suggested.

  • Q1-Q3 incorrect for readiness - see below highlighted. "The median values for the contemplativeness, readiness, and action taking subscales were 12 (Q1-Q3: 9-15, Min = 7, Max = 35), 21 (Q1-Q3: 914-26, Min = 4, Max = 20), and 24 (Q1-Q3: 18-29, Min = 8, Max = 40)"

Response: We have addressed the typographical error. It should have read 14-26

  • Table 2. The staff identity at LGBTQ2S. Should be "as"

Response: Thank you for flagging this. We have addressed the typographical error.